# Improvement in Chronic Low Back and Intermittent Chronic Neck Pain, Disability, and Improved Spine Parameters Using Chiropractic BioPhysics^®^ Rehabilitation After 5 Years of Failed Chiropractic Manipulation: A Case Report and 1-Year Follow-Up

**DOI:** 10.3390/healthcare13070814

**Published:** 2025-04-03

**Authors:** Katally Sanchez, Jason W. Haas, Paul A. Oakley, Deed E. Harrison

**Affiliations:** 1Private Practice, Aguadilla 00605, Puerto Rico; sanchezkatally@gmail.com; 2Chiropractic Biophysics NonProfit, Inc., Windsor, CO 80550, USA; 3Kinesiology and Health Science, York University, Toronto, ON M3J 1P3, Canada; docoakley.icc@gmail.com; 4Chiropractic BioPhysics NonProfit, Inc., Eagle, ID 83616, USA; drdeed@idealspine.com

**Keywords:** chronic low back pain, lumbar lordosis, cervical lordosis, chronic neck pain

## Abstract

Background/Objectives: We present a case documenting the successful treatment for a patient with chronic low back pain (CLBP), chronic neck pain (CNP), and decreased quality of life improving after conservative therapy. CLBP has been the leading cause of disability globally for the past few decades, resulting in decreased quality of life physically and emotionally. This case is important in the medical literature to add to studies reporting successful conservative treatment of CLBP and CNP. Triage, diagnosis, and understanding of economical and conservative therapeutics can benefit patients; providers as well as institutions and third party payors benefit from improved outcomes. Methods: A 39-year old male presented with severe CLBP who had experienced no long-term success with prior chiropractic spinal manipulative therapy (SMT). After symptoms began to worsen in spite of receiving SMT, the patient sought treatment for his pain, abnormal spine alignment, and poor sagittal alignment at a local spine facility. History and physical examination demonstrated altered spine and postural alignment including significant forward head posture and reduced cervical and lumbar lordosis and coronal plane abnormalities. Treatment consisted of a multi-modal regimen focused on strengthening postural muscles, specific spine manipulation directed toward abnormal full-spine alignment, and specific Mirror Image® traction aiming to improve spine integrity by realigning the spine toward a more normal position. The treatment consisted of 36 treatments over three months. All original tests and outcome measures were repeated following care. Results: Objective and subjective outcome measures, patient-reported outcomes, and radiographic mensuration demonstrated improvement at the conclusion of treatment and maintained at 1-year follow-up re-examination. Conclusions: This case demonstrates that the CBP^®^ orthopedic chiropractic treatment approach may represent an effective method to treat abnormal spinal alignment and posture. This study adds to the literature regarding conservative methods of treating spine pain and spinal disorders.

## 1. Introduction

This is a case documenting the changes in pain and disability in a patient suffering from low back and neck pain for several years. The patient received prior conservative chiropractic therapy but did not demonstrate measurable improvements subjectively or objectively until receiving a specific therapeutic approach; namely, the Chiropractic BioPhysics^®^ (CBP^®^) technique. Chronic low back pain (CLBP) and chronic neck pain (CNP) are major contributors to the global burden of disease (GBD) [1] and are frequently rated amongst the greatest causes of disability worldwide, with CLBP listed as first and CNP listed at fourth. It is estimated that annually, more than USD 40 billion dollars and more than USD 2000 per patient is spent on the treatment of low back pain in the US and globally [2]. This burden of suffering for the patient and the societal financial burden significantly contributes to years lived with disability (YLDs) and is an important research topic for the biomedical healthcare fields in order to elucidate effective strategies that offer safe, effective, and economical treatment protocols [3].

Abnormal spine and postural alignment are known contributors to chronic pain, and recent studies have shown that conservative therapies can reduce abnormal spine parameters, improve posture, and reduce pain following treatment [4,5,6,7,8,9]. Continued evaluation of biomechanical approaches for the assessment and treatment of CLBP is important, as emerging evidence is pointing to successful outcomes using such methods. Thus, the purpose of this case is to document the successful treatment of a patient with chronic spine pain by methods that have shown efficacy in improving pain and dysfunction by improving postural muscle strength and function, reducing abnormal postural loads and improving the normal configuration of the spine [5,6,7,8,9]. The (CBP^®^) protocol used on this patient will be outlined and described in detail in the methodology section. This case details the history and subjective complaints, intake evaluations including radiography and postural analysis, and the outcome measure questionnaires used at the initial, re-examination, and long-term follow-up exams.

## 2. Methods

### 2.1. Ethical Considerations, Patient History, Examination Results, and Subjective and Objective Findings

Institutional review board approval is not applicable due to the retrospective nature of this study, in accordance with Common Rule’s exemption 45 CFR 46.104. Informed consent and patient consent to publish were acquired prior to publication, the report does not contain any identifiable personal data, and thus the Declaration of Helsinki is not applicable for this case report. This analysis was conducted with a commitment to the utmost ethical integrity and data were used in a way that poses no risk to individuals and supports a broader understanding of spine rehabilitation and pain therapeutics.

A 38-year-old male suffering with constant, severe (7/10 on a numeric rating scale (NRS) of 0–10 where 0 is no pain and 10 is max pain) CLBP for over 5 years presented for treatment of his dysfunction and disability due to the ailment. The original cause of the low back pain was insidious and worsened over several years. Chronic intermittent neck pain was reported without a known cause. The pain would worsen with long days at work and with prolonged driving. The patient sought treatment with chiropractic spinal manipulation sporadically over five years with minimal short-term improvement in symptoms and disability with no long-term resolution. Due to the worsening of his condition, he sought out a spine rehabilitation facility in Mount Pleasant, South Carolina, USA specializing in conservative physical medicine methods for treating CBP.

The patient received a thorough history, physical examination, neurological and orthopedic testing, and subjective and objective assessment using outcome measures. Patient reported outcomes (PROs) were assessed for multiple regions of the spine and general health. The revised Oswestry low back pain disability index (RODI) [10] questionnaire was completed to determine how his CLBP pain affected his daily life and to assess self-rated LBP disability. His initial RODI score was 54% indicating severe disability due to low back pain. The patient completed a short-form 36 (SF-36) health-related quality of life (HRQoL) questionnaire to assess self-reported wellness [11]. The SF-36 outcome assessment provides scaled scores for eight different HRQoL domains on a scale of 0 to 100, where 0 is the lowest and 100 is the highest possible score. All of the initial SF-36 scores measured below normal indicating a significant impact of the chronic pain on the patient’s quality of life.

### 2.2. Postural Analysis and Radiographic Findings

Posture examination and analysis was performed using PostureScreen^®^ mobile posture analysis software (version 3.1.) (PostureCo, Inc., Trinity, FL, USA) [12,13] and revealed the following: in the sagittal plane, the patient stood with anterior head translation (+T_z_^H^), posterior thorax translation (−T_z_^T^), and anterior pelvic translation (+T_z_^P^) (Figure 1A).

Radiography of the spine was performed with the patient upright in neutral position. Anterior–posterior (AP) and lateral films were assessed using PostureRay^®^ digital radiographic mensuration software (Posture Co, Inc., Trinity, FL, USA) [14]. The program uses the Risser–Ferguson measurement for the AP views and the Harrison posterior tangent method (HPTM) of measurement for the lateral views. The program also compares the patient alignment of both segmental and global measurements to models of ideal spine parameters; prior studies have shown exceptionally high reliability using this system [15]. The sagittal spine absolute rotation angle (ARA) from the 2nd cervical vertebra to the 7th (ARA^C2–C7^) measured −18.1° (average is −34°, ideal is −42°, pain threshold is −20° [16]) (Figure 2, Figure 3, Figure 4 and Figure 5) and the AP cervical X-ray demonstrated a right lateral flexion angle relative to true vertical of the lower cervical and upper thoracic spine (cervicodorsal angle (CDA)) measuring 5.6° (ideal is 0° [16]) with a right translation of C2 with respect to T5 (T_X_^C2–T5^) measuring −17.2 mm (ideal is 0 mm). There was an increased mid-thoracic angle with a right-sided concavity from T1 to T12 (MTA^T1–T12^) of 9.1° (ideal is 0° [16]) with an increased translation at T8 apex of mid-thoracic angle with respect to T12 (+T_X_^T8–T12^) measuring 15.2 mm (ideal is 0 mm). There was a decreased sagittal curvature of the lumbar spine from L1 to L5 (ARA^L1–L5^) measuring −17.9° (ideal is −40°, average range is 35–45° [16]). The modified Ferguson pelvic radiograph demonstrated a sacral base unleveling in the frontal plane measuring −11.3 mm, being lower on the right. There was a lumbosacral angle from L1 to L5 with an L3 apex (LSA L1–L5) of −84.9° (ideal is 90°).

### 2.3. Treatment Protocols

Following the examination and analysis the patient agreed to participate in a multi-modal regimen to reduce the measured abnormal postural findings, reduce abnormal spine parameters, and reduce pain and dysfunction. The patient was treated 36 times in-office over 3 months [16]. The patient received manual spinal manipulative therapy (SMT) and CBP Mirror Image^®^ (MI^®^) adjustments using a chiropractic drop table and instrument adjusting using an Impulse^®^ high-velocity low-amplitude (HVLA) adjusting instrument (Neuromechanical Innovations, Chandler, AZ, USA) [17]. These MI^®^ adjustments involve positioning the patient in the mathematical opposite direction of the measured postural and spine abnormalities. Once the patient is properly positioned, a drop table mechanism or mechanical instrument is used to place the patient in the corrected posture.

MI^®^ traction was performed using a Universal Tractioning Systems^®^ (UTS^®^) total spine device (Universal Tractioning Systems, Inc., Las Vegas, NV, USA) with the patient in a semi-seated position on a stool with a cervical extension strap holding the skull and upper cervical spine in extension with a counter stress pulling from posterior to anterior through the mid cervical spine. The traction was applied for 5–12 min with 12 lbs in the posterior and 25 lbs in the anterior. Further, a posterior-to-anterior (PA) static pull using a comfortable strap on a pully system was positioned at L3–L4 through the disc plane line. Additionally, an AP static pull with a comfortable padded strap at T6-T8 to stabilize the thorax, while a PA dynamic weighted cervical pull at the mid-cervical spine is applied with a small, circular padded comfortable strap angled at the mid-cervical spine (Figure 6). Further, a posterior and superior static distraction pull to the head and neck using a padded occipital and chin harness. The patient also performed seated UTS^®^ MI^®^ traction with a left head translation (+T_x_^H^) pull and a right thoracic counter-stress [18].

MI^®^ exercises were performed including +T_x_^H^, left thoracic rotation or lateral flexion (−R_z_^T^), and extension exercises for strengthening the lumbar spine musculature. The patient was prescribed daily home care consisting of MI^®^ traction and exercises held for no longer than 15 s and worked up to 50 repetitions a day. The exercises and traction were completed at increasing tolerance to soreness using a Pro-Lordotic Neck Exerciser^TM^ (Circular traction, LLC. Huntington Beach, CA, USA) (Figure 7) [19] and cervical and lumbar Dennerolls^®^ (Denneroll™ cervical traction orthotic (DCTO) and Denneroll^TM^ lumbar traction orthotic (DLTO)) to create cervical and lumbar extension moments based on previously published average, ideal, and pain threshold models [14,15] The DCTO and DLTO have a very specific peak that is placed at the area of the spine that requires extension moment as measured on the PostureRay^®^ sagittal spine profile mensuration parameters. The patient was also prescribed a 12 mm cork full foot lift to be worn in the right footwear to correct for a right short anatomical leg length inequality (ALLI) that was measure on the AP modified Fergusion radiograph for ALLI assessment. All exercises were to patient tolerance, beginning with few repetitions and increasing to as many as 50 repetitions per day with an intensity and duration beginning at 3–5 s and increasing to as sustained hold of no more than 15 s. Similar traction protocols were followed with the patient beginning the traction for 2–5 min per session depending on tolerance and increasing to up to 15 min of traction with progressively increased tension in the stresses on the spine in the MI^®^ position.

## 3. Results and 1-Year Follow-Up

Post-treatment posture analysis showed improved posture (Figure 1A–C). Post-treatment radiographic examination revealed the following: improved ARA^C2–C7^ measuring −29.4° (vs. −18.1°); rotation around the *z*-axis of the thorax (R_Z_^T5^) measured 1.8° (vs. normal 0°); improved -T_x_^C2–T5^ measuring −5.7 mm (vs. −17.2 mm); improved MTA T1–T12 measuring 2.1° (vs. 9.1°); improved +T_x_^T8–T12^ measuring 3.5 mm (vs. 15.2 mm); improved ARA^L1–L5^ measuring −25.1° (vs. −17.9°); improved sacral base unleveling in the frontal plane measuring −1.0 mm low on the left (vs. −11.3 mm low on the right); and improved LSA L1–L5 of −88.0° (vs. −84.9°) (Figure 2B, Figure 3B, Figure 4B and Figure 5B). Post-treatment RODI score was 12% (vs. 54%), indicating minimal disability. All post-treatment SF-36 scores showed improvements (Table 1). One-year follow-up posture analysis showed a maintenance of the improved posture. One-year follow-up radiographic examination revealed maintained sagittal balance and coronal spinal alignment correction improvements (Figure 2C, Figure 3C, Figure 4C and Figure 5C). One-year follow-up RODI score was 2% indicating minimal or resolved disability from baseline (54%). Post-treatment SF-36 scores showed maintained or further improved HRQoL measures reported by the patient. Long term follow-up found minimal forward head posture on the lateral posture photograph, a slight return to baseline on the A-P cervical radiograph with a right head translation measuring 7 mm. Lateral cervical radiograph assessment at long-term found the lordosis to be well maintained at 34° ARA with minimal C2-vertical anterior head translation of 6 mm. lateral lumbar radiograph showed a slight loss of lordosis at follow-up of 19°. All subjective initial symptoms were reported to be resolved at long-term follow-up. Long term follow-up SF-36 scores were the same as post-treatment with the exception of vitality, which was slightly improved (Table 1). There were no positive orthopedic or neurological tests at follow-up. The patient continued to use the ProLordotic neck exerciser at home 1–3 times per week for up to 10 min (Figure 7).

## 4. Discussion

This case documents the successful treatment of a male who suffered from chronic spine pain and significant disability. After CBP^®^ conservative spine rehabilitation, there was an increase in cervical and lumbar lordosis, decrease in lateral head and thoracic translation; all the postural improvements corresponded with improvements in pain, disability, and function after only 3 months. The improved posture and symptom reductions were maintained at a long-term follow-up.

The treatment outcomes for the present case were consistent with prior reported case reports [19], cohort and case series [20], randomized controlled trials [21,22,23,24,25], systematic reviews of the literature, and other prior investigations [26,27,28,29]. This protocol is designed to target abnormal postural permutations by addressing musculature asymmetries. Based on the abnormal postural and radiographic findings, a specific program of MI^®^ exercises that use repetition and sustained contraction to improve coronal and sagittal balance. The exercises begin with minimal intensity and progress to contractions being held for a longer period of time as well and increased repetitions. The presence of abnormal postural positions causing musculature asymmetries has been reported for many decades [30].

SMT has been used for a very long time to treat spine pain with variable results. General spinal manipulation is known to increase range of motion and decrease pain temporarily; however, no studies have shown consistent long-term improvement in subjective and objective outcomes for SMT. This is consistent with our patient who received traditional SMT from a chiropractor without long-term reduction in symptoms [31]. The MI^®^ SMT used in this study was different due to the fact that each manipulation is based on moving the patient toward improved sagittal and coronal balance. Improving postural abnormalities has previously been shown to reduce abnormal loads on soft tissues such as disc and spine ligaments as well and improve latency and amplitude in the central nervous system [32]. It has been theorized that abnormal postures increase energy expenditure in the spinal musculature and reduce efficiency in the nervous system centers involved with postural control [7,33,34]. MI^®^ SMT involves placing the patient in the mathematical opposite of the measured spinal abnormalities and introducing a force via a drop-table mechanism or a HVLA instrument to improve neuroplastic changes in postural control areas in the CNS [6].

Postural MI^®^ traction is designed to influence viscoelastic tissues such as the spinal ligaments and intervertebral discs that do not respond to rapid impulses from SMT but require time-dependent forces to cause creep in the direction of normal spine curvatures in the sagittal plane and toward a straight spine in the coronal plane [35]. The traction forces are applied according to the measured abnormal spine parameters and involve as-comfortable-as-possible patient positioning to hold the spine in a closer-to-normal position and slowly increased traction intensity and time so as to stretch the viscoelastic structures towards a more neutral and balanced posture. The results of this case is consistent with prior studies of the MI^®^ traction and were found in conjunction with the positive subjective outcome measures and PROs following treatment and correction was sustained at long-term follow-up. These methods of improving sagittal balance and cervical and lumbar lordosis have been shown to be repeatable and reliable in inducing increases in lumbar and cervical lordosis toward a more ideal angulation in both conservative and surgical studies [5,6,8,14,15,16,17,18,19,22,23].

Radiography in the treatment of spine conditions has a long reported history, however, there are those who claim that radiography is dangerous [36,37,38]. Radiography has historically been a fear-inducing topic, due to concerns surrounding the genetic damage induced, but this traditional idea is not currently supported by data [39,40,41] that show the hormetic effects that occur in the body. That is, although damage occurs initially, the body is an adaptive organism, whereby any initial damage is repaired. Thus, low-dose exposures including spinal X-rays do not present a net effect of damage. Therefore, the clinical effectiveness of radiography is supported as only a net benefit results as there are no risks from a risk-to-benefit perspective [35,39,41]. Having reliable and affordable treatment options is desirable across multiple levels of administrative and provider care. This case demonstrating the effectiveness of CBP^®^ care offers an alternative treatment protocol, with reliable and economical results.

Further studies are necessary to warrant the inclusion of these protocols in governing bodies’ recommendations and best practice guidelines for the treatment of these and other conditions. Limitations of this study are the reportage of results on only one patient; however, these results are consistent with the prior literature. Limitations also include the relatively short long-term follow-up (1 year); longer-term (5-year or 10-year) follow-up studies would provide better confirmation of long-term benefits. Additionally, the subjective nature of PROs could be improved with additional functional testing and longer-term follow-up of the findings.

## 5. Conclusions

The results of this single case indicates that CBP^®^ corrective chiropractic care may be an effective method to treat abnormal spinal alignment parameters and abnormal posture that contributes to CLBP, CNP, and disability causing decreased HRQoL and abnormal PROs. A simple multi-modal conservative treatment protocol could reduce financial burden for treating low back and neck pain and reduce risks associated with pharmacological and more invasive interventions. Improvement in spinal alignment and posture may result in long-term reduction or resolution of cLBP and improved HRQoL. This case shows the need for more conservative research involving spinal biomechanics and rehabilitation in patients with spine pain.

## Figures and Tables

**Figure 1 healthcare-13-00814-f001:**
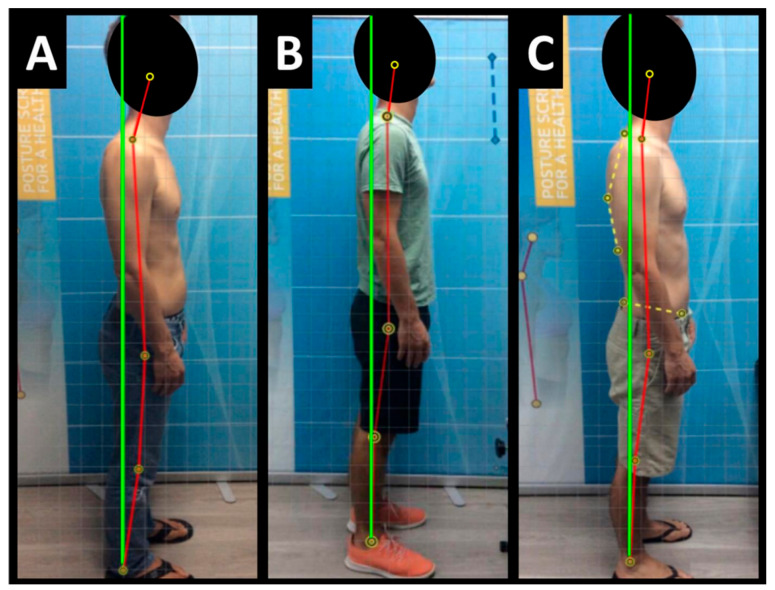
(**A**–**C**) Pre-treatment, post-treatment, and 1-year follow-up sagittal posture. Image Features: The green line represents a normal, ideal sagittal posture. The red line represents the actual sagittal posture of the patient. The yellow circles with the black ring inside are anatomical landmarks made to analyze the patient’s posture.

**Figure 2 healthcare-13-00814-f002:**
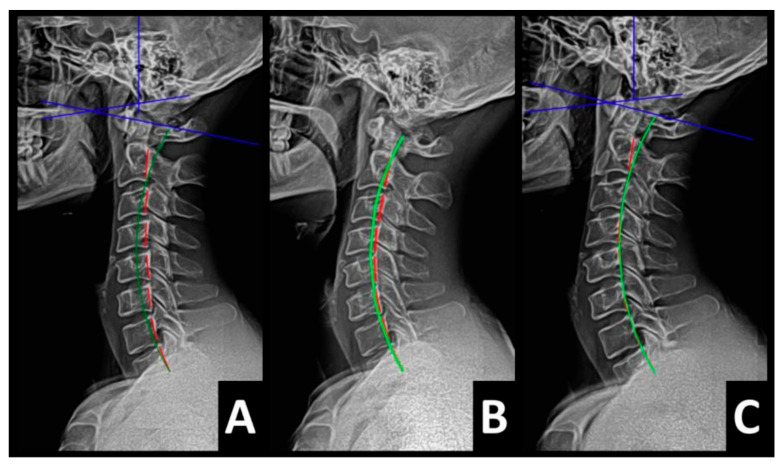
(**A**–**C**) Pre-treatment, post-treatment, and 1-year follow-up neutral lateral cervical radiographs. Image Features: The green line represents a normal, ideal sagittal cervical spinal alignment. The red line represents the actual posterior tangent lines of the C2–1 vertebrae.

**Figure 3 healthcare-13-00814-f003:**
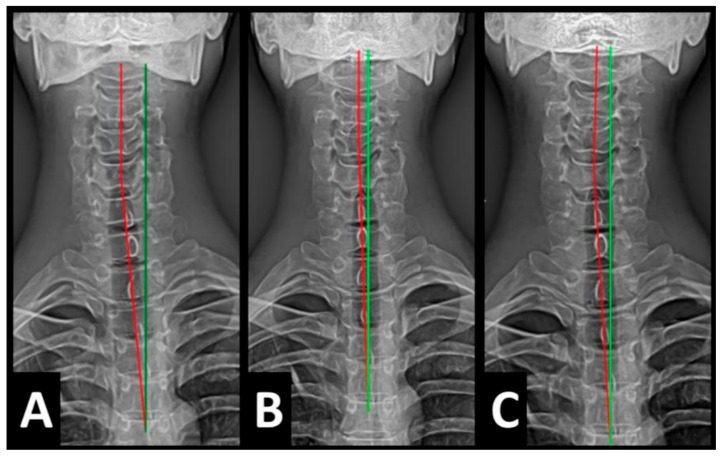
(**A**–**C**) Pre-treatment, post-treatment, and 1-year follow-up AP cervical radiographs. Image Features: The green line represents a normal, ideal frontal cervical spinal alignment. The red line represents the actual frontal cervical alignment of the C1–T5 vertebrae. The right side of the radiographs are the left side of the patient.

**Figure 4 healthcare-13-00814-f004:**
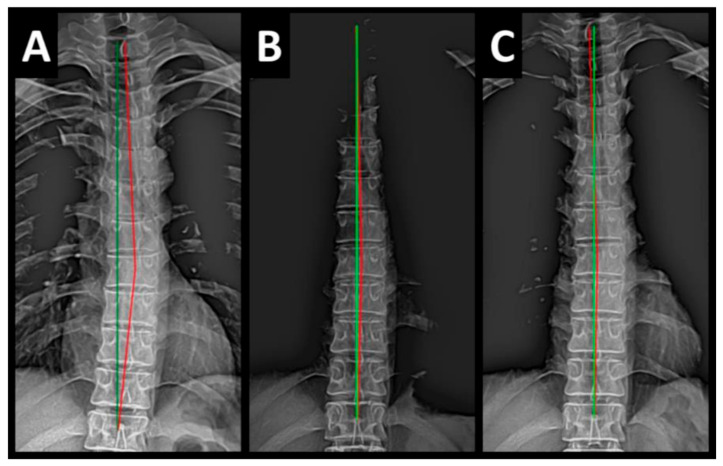
(**A**–**C**) Pre-treatment, post-treatment, and 1-year follow-up AP thoracic radiographs. Image Features: The green line represents a normal, ideal frontal thoracic spinal alignment. The red line represents the actual frontal alignment of the T1–T12 vertebrae. The right side of the radiographs are the left side of the patient.

**Figure 5 healthcare-13-00814-f005:**
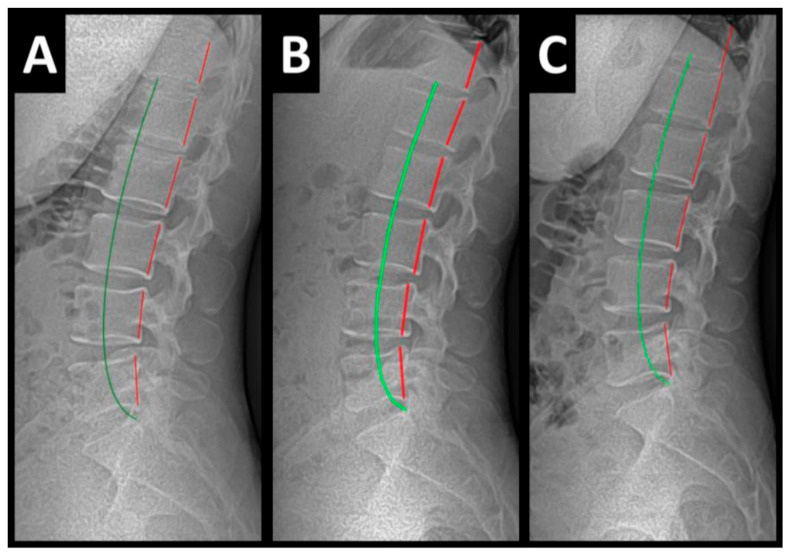
(**A**–**C**) Pre-treatment, post-treatment, and 1-year follow-up lateral lumbar radiographs. Image Features: The green line represents a normal, ideal sagittal lumbar spinal alignment. The red line represents the actual lumbar alignment of the T12–L5 vertebrae.

**Figure 6 healthcare-13-00814-f006:**
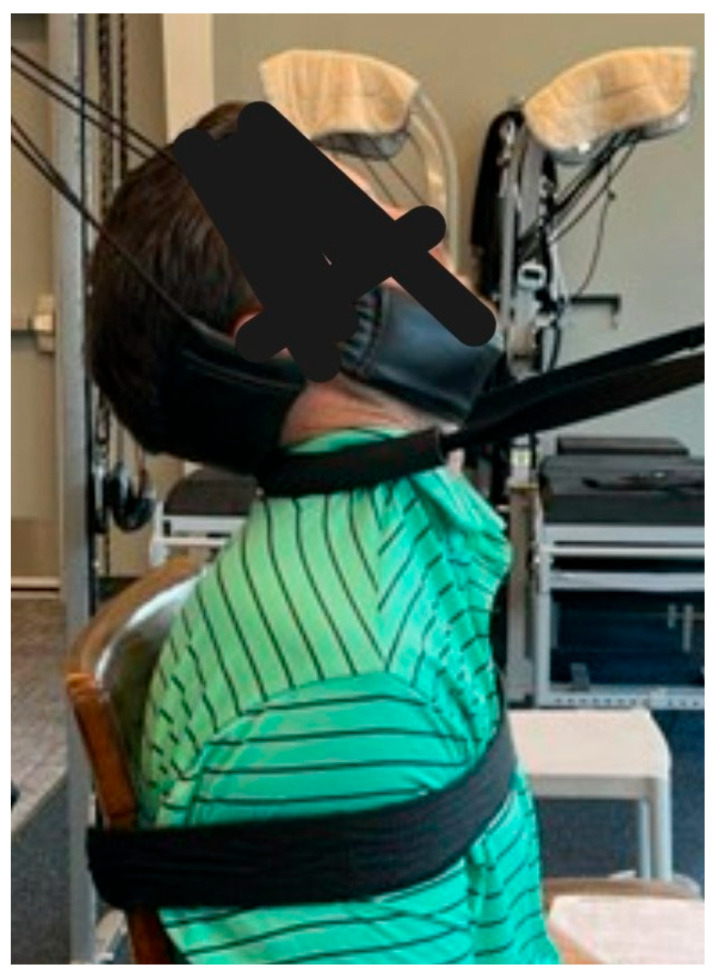
Cervical extension Mirror Image^®^ traction. Image description: A sample image of a patient undergoing Mirror Image^®^ traction with a comfortable padded strap on the jaw and occiput and a counter strap through the mid cervical spine in a posterior–anterior and cephalad direction. The front load was 25 lbs and the posterior load was static with a fixed pulley.

**Figure 7 healthcare-13-00814-f007:**
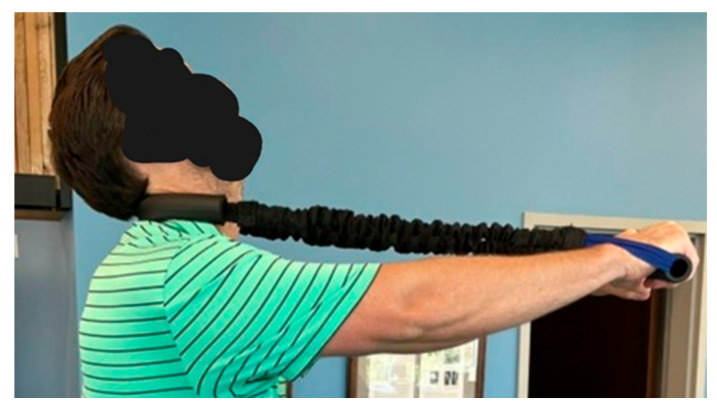
The ProLordotic^TM^ neck exercise device. Image Description: An elastic strap through the mid cervical spine is lengthened against extension resistance of the cervical paraspinal muscles. The contraction is held for 15 s and performed for no longer than 10 min.

**Table 1 healthcare-13-00814-t001:** Pre-treatment, post-treatment, and 1-year follow-up results of the SF-36 Health Status Questionnaire.

SF-36 Quality of Life Scales	Normative Mean Scale Scores	Pre-CBP^®^ Treatment Exam 05/31/2018	Post-CBP^®^ Treatment Exam 01/09/2019	Post-CBP^®^ Follow-Up Exam 08/07/2019
PF	72.0	25.0	90.0	90.0
RP	81.0	0.0	100.0	100.0
RE	81.0	0.0	100.0	100.0
VT	61.0	45.0	60.0	65.0
MH	81.0	32.0	80.0	80.0
SF	83.0	12.5	87.5	90.0
BP	75.0	22.5	77.5	77.5
GH	72.0	50.0	80.0	80.0
ΔH	84.0	45.0	100.0	100.0

Note: PF: physical functioning; RP: role limitations due to physical health problems; RE: role limitations due to personal or emotional problems; VT: vitality; MH: mental health; SF: social functioning; BP: bodily pain; GH: general health; ΔH: change in health status. Grey indicates normative mean values.

## Data Availability

These data were derived from the following resources available in the public domain. The authors declare that this literature review is not based on original data.

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
