# Peer review of "Improvement in Chronic Low Back and Intermittent Chronic Neck Pain, Disability, and Improved Spine Parameters Using Chiropractic BioPhysics® Rehabilitation After 5 Years of Failed Chiropractic Manipulation: A Case Report and 1-Year Follow-Up"

_healthcare, 2025, doi:10.3390/healthcare13070814_

Round 1

Reviewer 1 Report

Comments and Suggestions for Authors

Dear Authors,

I appreciate the opportunity to review your manuscript. I find the article interesting, well-written, and providing valuable information. However, to meet the journal's standards, I suggest the following minor revisions:

  1. Introduction:

    1. At the beginning of the introduction please avoid speaking in a personal way. 
    2. Modify the sentence "The (CBP®) protocol used on this patient will be outlined and described in detail." To clarify, the treatment protocol will be described in detail in the methodology section.

  2. Methods:

    1. The characterization data of the subject's condition and the tables and figures are better suited for the results section. Please don't include them in the methodology; instead, explain only the tools used in the evaluations and the applied treatment.

    2. Please include information on the ethical treatment of data, ethics committee approval, and adherence to the principles of the Declaration of Helsinki.

  3. Figures:

    1. In Figures 2A, 3A, 4A, and 5A, edit the green line to ensure it is as visible as in Figures B and C. 

    2. Figure 6: For greater accuracy, please include the lumbar straps in the image.
  4. Treatment Protocols:

    • In explaining MI(R) traction, correct the spelling of the word "extension" and the word "othotic" as well as the "DLTO" abbreviation..

    • In the same section, the anterior load is described as 20 pounds, while it is 25 pounds in the image. Similarly, the posterior traction load is described as 12 pounds, but it appears secured with a strap in the picture. We understand this refers to cervical traction on one side and lumbar traction on the other. Could you clarify this in both cases?

  5. Results: Include the quantitative results of all reevaluations at the one-year follow-up compared to the pre- and post-treatment evaluations. To help you understand and track patient progress, you can present this data in table format.

  6. Discussion: The paragraph "Chronic spine pain including CLBP and CNP are a global epidemic and are consistently rated as the 1st and 4th leading causes of disability globally [1]. These conditions produce a significant burden for health care providers, patients, and societal financial burden [2]. Having repeatable, reliable and consistent protocols for the diagnosis and treatment of spine conditions is important to improve the assessment and recommendations for this patient population which grows larger each year [3]." is a paraphrased version of the final part of the first paragraph in the introduction. In the discussion, results should be framed concerning existing evidence. Please remove this paragraph or justify why it is repeated.

  7. References: Carefully review the bibliography: standardize the way page numbers are presented in references 2, 6, 7, 11, 14, 15, 16. Additionally, references 31 and 32 are duplicated. 

Author Response

Thank you for your review of our manuscript. The following corrections were made:

Introduction:

1) Corrected and revised

2) Modified and corrected

Methods:

1) Table was moved to the Results as recommended

2) ethics statement was made with reference to IRB exemption

Figures:

1) The green line is not able to be enhanced.

2) There are no lumbar straps in the image

Treatment Protocols:

All suggested corrections were made

Results:

The table 1 was moved to the results section as recommended.

Discussion:

Redundant paragraph was removed

References:

All references were corrected for format and page numbers and repeated reference was corrected.

Thank you again for the review of our manuscript.

Sincerely, 

The Authors

Reviewer 2 Report

Comments and Suggestions for Authors

Dear Authors,

it is my pleasure to review your study.

Article titled "Improvement in Chronic Low Back and Intermittent Chronic Neck Pain, Disability, and Improved Spine Parameters Using Chiropractic BioPhysics® Rehabilitation After 5 Years of Failed Chiropractic Manipulation: A Case Report and 1-Year Follow-Up" raises an interesting topic but I have a lot of doubts.

-Please clearly define the diagnosis, what exactly caused the patient's symptoms? CLBP can be caused by many pathologies. Before starting treatment, it is necessary to determine what is the primary cause of the symptoms.

-If the patient was treated for several years without any effect, then most likely the wrong diagnosis was made. Was an MRI or CT scan performed?

-In Figure 1, what is the basis for determining that the green line represents the correct image? By what criteria?

-In figure no. 1, the postural photo should not be taken in shoes, it should be taken barefoot, moreover, in each photo the patient has different shoes. This figure should be removed.

-In my opinion, the radiographic analysis in Figure 1 is medically unreliable and should be removed from the article. The methodology of the study, especially in scientific articles, plays a key role.

-In Figure 2, what is the basis for determining that the green line represents the correct image? By what criteria?

-In Figure 3-5, what is the basis for determining that the green line represents the correct image? By what criteria?

-The authors state many irregularities in the patient's examinations, however, it was not specified on the basis of which criteria, according to which guidelines, insufficient diagnostic imaging was performed, and the medical examination was not precisely described. Assessment on various scales is not sufficient in assessing the patient's clinical condition, assessment on scales is necessary to assess the effectiveness of therapy and publication of results in the literature.

-In the discussion, attention is drawn to the excess of citations, e.g. "The treatment outcomes for the present case were consistent with previously reported case reports [19], cohort and case series [20], randomized controlled trials [21,22,23,24,25] , systematic reviews of the literature and other prior investigations [26,27,28,29]."

-Due to methodological irregularities and the lack of precise diagnosis, the presented conclusions are inadequate.

Author Response

Thank you for the review of our manuscript.

Respectfully, we disagree with the reviewers comments and will not be making any changes based upon this review.  We feel the reviewer has made numerous false accusations which may indicate bias.

Author response file is attached.

Please see the changes made by the higher quality recommendations of Reviewer 1.

Sincerely, 

The authors

Reviewer 3 Report

Comments and Suggestions for Authors This article, titled "Improvement in Chronic Low Back and Intermittent Chronic Neck Pain, Disability, and Improved Spine Parameters Using Chiropractic BioPhysics® Rehabilitation After 5 Years of Failed Chiropractic Manipulation: A Case Report and 1-Year Follow-Up," offers a detailed account of the successful management of chronic low back pain (CLBP) and chronic neck pain (CNP) using Chiropractic BioPhysics® (CBP®) rehabilitation techniques. From an orthopedic standpoint, the article provides valuable insights while also presenting certain limitations. One of the article's notable strengths lies in its clinical significance. By focusing on CLBP and CNP, conditions that are major causes of disability worldwide, the study addresses a critical issue in orthopedics and rehabilitation medicine. The successful treatment and the long-term follow-up not only highlight the potential benefits of CBP® methods but also suggest that these methods could be a valuable addition to conservative treatment protocols for spine-related pain and dysfunction. The authors conducted a comprehensive assessment, utilizing a combination of patient-reported outcomes (PROs) such as the Oswestry Low Back Pain Disability Index (RODI) and the SF-36 Health-Related Quality of Life questionnaire, along with objective radiographic measurements. This multifaceted approach provides a well-rounded evaluation of the patient's condition and the treatment outcomes. The article also stands out for its detailed treatment protocol. The authors meticulously described the CBP® treatment process, including manual spinal manipulative therapy (SMT), Mirror Image® adjustments, traction, and home exercises. This level of detail is crucial for the reproducibility of the treatment methods, making it a valuable resource for further research and clinical application. The use of radiographic measurements to assess spinal alignment and posture is another strong point of the study. By employing the PostureRay® digital radiographic mensuration software, which is known for its high reliability, the authors were able to document improvements in spinal parameters such as cervical and lumbar lordosis, providing objective evidence of the treatment's effectiveness. The 1-year follow-up examination is a significant strength of the study, demonstrating the sustained benefits of the CBP® treatment. The maintenance of improved posture and spinal alignment, along with the continued reduction in pain and disability, underscores the potential long-term efficacy of this approach. However, the study is not without its limitations. The single-case design restricts the generalizability of the findings, and while the results are promising, further research involving larger sample sizes and controlled trials is necessary to validate the effectiveness of CBP® methods for the treatment of CLBP and CNP. The short long-term follow-up is another limitation. Although a 1-year follow-up is provided, the long-term effects of the treatment beyond this period are not assessed. Given that chronic spine conditions often require long-term management, a longer follow-up period (e.g., 5-10 years) would offer more comprehensive data on the durability of the treatment effects. The absence of a control group also makes it difficult to determine whether the observed improvements are specifically attributable to the CBP® treatment or to other factors such as natural recovery or the placebo effect. Future studies should include control groups to better isolate the treatment effects. Additionally, the use of radiographic imaging, while necessary for assessing spinal alignment, raises concerns about radiation exposure. Although the authors argue that low-dose exposures do not present significant risks, the potential cumulative effects of multiple radiographic assessments should be considered, especially in long-term studies. While the use of PROs is valuable, the subjective nature of these measures introduces potential bias. Objective measures, such as functional tests and biomechanical assessments, could complement the PROs and provide a more objective evaluation of the patient's functional status.

Author Response

Thank you for the review of our manuscript.

We appreciate the comments and recommendations made in your review and have made some modifications to the discussion section of the study with expansion of the limitations and clarification of the findings.

Thank you

Round 2

Reviewer 2 Report

Comments and Suggestions for Authors

Dear Authors,

the cause of CLBP may be different. Therefore, precise diagnosis is crucial. Especially in the context of scientific research or publication of articles. LBP may be caused by, for example, a tumor, discopathy, or facet syndrome.

MRI and CT diagnostics are not standardly performed, but for the purpose of precise assessment and exclusion of serious pathologies, it is often necessary. Especially in scientific research, so as not to raise doubts.

The test for assessing cervical lordosis is an X-ray. Postural X-ray should be performed barefoot, not in shoes. This is a diagnostic error.

The authors did not make any changes, which significantly reduces the scientific quality of the article. First of all, science should be based on EBM.

Author Response

  • Please clearly define the diagnosis, what exactly caused the patient's symptoms? CLBP can be caused by many pathologies. Before starting treatment, it is necessary to determine what is the primary cause of the symptoms.

Reply- The diagnosis was chronic low back pain and neck pain (M54.2 and M54.9). The authors choose not to include the ICD-10 codes, as these are specific to the US and can confuse other providers who are not familiar with the ICD-10 codes. Abnormal biomechanics of the spine and posture have long been known to cause abnormal loading of the pain sensitive neural tissue and receptors found in the disc, facets, and intervertebral tissues. 

  • If the patient was treated for several years without any effect, then most likely the wrong diagnosis was made. Was an MRI or CT scan performed?

Reply- There was no MRI or CT performed/available and is not indicated for simple mechanical neck and low back pain which responded well to conservative care.

  • In Figure 1, what is the basis for determining that the green line represents the correct image? By what criteria?

Reply- Cervical and lumbar lordosis and vertical axis lines have long been established in the medical literature.  See references 5, 6, 8, 14, 15, 16, 18, 19, 20, 25, 26, 27, 28

Here are additional references regarding normal biomechanics: Betz JW, Lightstone DF, Oakley PA, Haas JW, Moustafa IM, Harrison DE. Reliability of the
Biomechanical Assessment of the Sagittal Lumbar Spine and Pelvis on Radiographs Used in
Clinical Practice: A Systematic Review of the Literature. J Clin Med. 2024 Aug 8;13(16):4650.
doi: 10.3390/jcm13164650.

Harrison DD, Janik TJ, Troyanovich SJ, Holland B. Comparisons of Lordotic Cervical Spine Curvatures to a Theoretical Ideal Model of the Static Sagittal Cervical Spine. Spine 1996;21(6):667-675. 

Harrison DD, Janik TJ, Troyanovich SJ, Harrison DE, Colloca CJ. Evaluations of the Assumptions Used to Derive an Ideal Normal Cervical Spine Model. J Manipulative Physiol Ther 1997; 20(4): 246-256.

Troyanovich SJ, Cailliet R, Janik TJ, Harrison DD, Harrison DE. Radiographic Mensuration Characteristics of the Sagittal Lumbar Spine From A Normal Population with a Method to Synthesize Prior Studies of Lordosis. J Spinal Disord 1997;10(5): 380-386

Janik TJ, Harrison DD, Cailliet R, Troyanovich SJ, Harrison DE. Can the Sagittal Lumbar Curvature be Closely Approximated by an Ellipse? J Orthop Res 1998; 16(6): 766-770

Harrison DE, Harrison DD, Janik TJ, Jones EW, Cailliet R, Normand M. Comparison of Axial and Flexural Stresses in Lordosis and Three Buckled Modes in the Cervical Spine. Clin Biomech 2001; 16(4): 276-284.

Harrison DD, Jones EW, Janik TJ, Harrison DE. Evaluation of Flexural Stresses in the Vertebral body Cortex and Trabecular Bone in Three Cervical Configurations with an Elliptical Shell Model. J Manipulative Physiol Ther 2002; 25(6): 391-401.

Harrison DE, Harrison DD, Janik TJ, Cailliet R, Haas JW. Do alterations in vertebral and disc dimensions affect an elliptical model of the thoracic kyphosis? Spine 2003; 28(5): 463-469

Keller TS, Colloca CJ, Harrison DE, Harrison DD, Janik TJ. Influence of spine morphology on intervertebral disc loads and stresses in asymptomatic adults: Implications for the Ideal Spine. Spine Journal 2005; 5:297-305.

Harrison DE, Haas JW, Harrison DD, Janik TJ, Holland B. Do Sagittal Plane Anatomical Variations (Angulation) of the Cervical Facets and C2 Odontoid Affect the Geometrical Configuration of the Cervical Lordosis? Results from Digitizing Lateral Cervical Radiographs in 252 neck pain subjects. Clin Anat 2005; 18:104-111.

Harrison DE, Harrison DD, Janik TJ, Holland B, Siskin L. Slight Head Extension: Does it Reverse the Cervical Curve? Euro Spine J 2001; 10: 149-153.

  • In figure no. 1, the postural photo should not be taken in shoes, it should be taken barefoot, moreover, in each photo the patient has different shoes. This figure should be removed.

Reply- The figure contains valid information for the reader and will not be removed. This is a retrospective case and no past figures will be doctored.

  • In my opinion, the radiographic analysis in Figure 1 is medically unreliable and should be removed from the article. The methodology of the study, especially in scientific articles, plays a key role.

Reply- Thank you for your opinion.  The authors kindly disagree and the photo will not be removed. and the reviewers comment demonstrates a fundamental lack of knowledge of the current state of the medical literature regarding normal and abnormal biomechanics.  Please read all of the above references to better understand how normal vs abnormal spine configurations are measured and treated. Reliability of the methods used in this case report have been exhaustively studied and previously reported in the medical literature. See Above references.

  • In Figure 2, what is the basis for determining that the green line represents the correct image? By what criteria?

Reply- Cervical and lumbar lordosis and vertical axis lines have long been established in the medical literature.  See references 5, 6, 8, 14, 15, 16, 18, 19, 20, 25, 26, 27, 28

Additional references: Betz JW, Lightstone DF, Oakley PA, Haas JW, Moustafa IM, Harrison DE. Reliability of the
Biomechanical Assessment of the Sagittal Lumbar Spine and Pelvis on Radiographs Used in
Clinical Practice: A Systematic Review of the Literature. J Clin Med. 2024 Aug 8;13(16):4650.
doi: 10.3390/jcm13164650.

Harrison DD, Janik TJ, Troyanovich SJ, Holland B. Comparisons of Lordotic Cervical Spine Curvatures to a Theoretical Ideal Model of the Static Sagittal Cervical Spine. Spine 1996;21(6):667-675. 

Harrison DD, Janik TJ, Troyanovich SJ, Harrison DE, Colloca CJ. Evaluations of the Assumptions Used to Derive an Ideal Normal Cervical Spine Model. J Manipulative Physiol Ther 1997; 20(4): 246-256.

Troyanovich SJ, Cailliet R, Janik TJ, Harrison DD, Harrison DE. Radiographic Mensuration Characteristics of the Sagittal Lumbar Spine From A Normal Population with a Method to Synthesize Prior Studies of Lordosis. J Spinal Disord 1997;10(5): 380-386

Janik TJ, Harrison DD, Cailliet R, Troyanovich SJ, Harrison DE. Can the Sagittal Lumbar Curvature be Closely Approximated by an Ellipse? J Orthop Res 1998; 16(6): 766-770

Harrison DE, Harrison DD, Janik TJ, Jones EW, Cailliet R, Normand M. Comparison of Axial and Flexural Stresses in Lordosis and Three Buckled Modes in the Cervical Spine. Clin Biomech 2001; 16(4): 276-284.

Harrison DD, Jones EW, Janik TJ, Harrison DE. Evaluation of Flexural Stresses in the Vertebral body Cortex and Trabecular Bone in Three Cervical Configurations with an Elliptical Shell Model. J Manipulative Physiol Ther 2002; 25(6): 391-401.

Harrison DE, Harrison DD, Janik TJ, Cailliet R, Haas JW. Do alterations in vertebral and disc dimensions affect an elliptical model of the thoracic kyphosis? Spine 2003; 28(5): 463-469

Keller TS, Colloca CJ, Harrison DE, Harrison DD, Janik TJ. Influence of spine morphology on intervertebral disc loads and stresses in asymptomatic adults: Implications for the Ideal Spine. Spine Journal 2005; 5:297-305.

Harrison DE, Haas JW, Harrison DD, Janik TJ, Holland B. Do Sagittal Plane Anatomical Variations (Angulation) of the Cervical Facets and C2 Odontoid Affect the Geometrical Configuration of the Cervical Lordosis? Results from Digitizing Lateral Cervical Radiographs in 252 neck pain subjects. Clin Anat 2005; 18:104-111.

Harrison DE, Harrison DD, Janik TJ, Holland B, Siskin L. Slight Head Extension: Does it Reverse the Cervical Curve? Euro Spine J 2001; 10: 149-153.

  • In Figure 3-5, what is the basis for determining that the green line represents the correct image? By what criteria?

Reply- Cervical and lumbar lordosis and vertical axis lines have long been established in the medical literature.  See references 5, 6, 8, 14, 15, 16, 18, 19, 20, 25, 26, 27, 28

Additional references: Betz JW, Lightstone DF, Oakley PA, Haas JW, Moustafa IM, Harrison DE. Reliability of the
Biomechanical Assessment of the Sagittal Lumbar Spine and Pelvis on Radiographs Used in
Clinical Practice: A Systematic Review of the Literature. J Clin Med. 2024 Aug 8;13(16):4650.
doi: 10.3390/jcm13164650.

Harrison DD, Janik TJ, Troyanovich SJ, Holland B. Comparisons of Lordotic Cervical Spine Curvatures to a Theoretical Ideal Model of the Static Sagittal Cervical Spine. Spine 1996;21(6):667-675. 

Harrison DD, Janik TJ, Troyanovich SJ, Harrison DE, Colloca CJ. Evaluations of the Assumptions Used to Derive an Ideal Normal Cervical Spine Model. J Manipulative Physiol Ther 1997; 20(4): 246-256.

Troyanovich SJ, Cailliet R, Janik TJ, Harrison DD, Harrison DE. Radiographic Mensuration Characteristics of the Sagittal Lumbar Spine From A Normal Population with a Method to Synthesize Prior Studies of Lordosis. J Spinal Disord 1997;10(5): 380-386

Janik TJ, Harrison DD, Cailliet R, Troyanovich SJ, Harrison DE. Can the Sagittal Lumbar Curvature be Closely Approximated by an Ellipse? J Orthop Res 1998; 16(6): 766-770

Harrison DE, Harrison DD, Janik TJ, Jones EW, Cailliet R, Normand M. Comparison of Axial and Flexural Stresses in Lordosis and Three Buckled Modes in the Cervical Spine. Clin Biomech 2001; 16(4): 276-284.

Harrison DD, Jones EW, Janik TJ, Harrison DE. Evaluation of Flexural Stresses in the Vertebral body Cortex and Trabecular Bone in Three Cervical Configurations with an Elliptical Shell Model. J Manipulative Physiol Ther 2002; 25(6): 391-401.

Harrison DE, Harrison DD, Janik TJ, Cailliet R, Haas JW. Do alterations in vertebral and disc dimensions affect an elliptical model of the thoracic kyphosis? Spine 2003; 28(5): 463-469

Keller TS, Colloca CJ, Harrison DE, Harrison DD, Janik TJ. Influence of spine morphology on intervertebral disc loads and stresses in asymptomatic adults: Implications for the Ideal Spine. Spine Journal 2005; 5:297-305.

Harrison DE, Haas JW, Harrison DD, Janik TJ, Holland B. Do Sagittal Plane Anatomical Variations (Angulation) of the Cervical Facets and C2 Odontoid Affect the Geometrical Configuration of the Cervical Lordosis? Results from Digitizing Lateral Cervical Radiographs in 252 neck pain subjects. Clin Anat 2005; 18:104-111.

Harrison DE, Harrison DD, Janik TJ, Holland B, Siskin L. Slight Head Extension: Does it Reverse the Cervical Curve? Euro Spine J 2001; 10: 149-153.

  • The authors state many irregularities in the patient's examinations, however, it was not specified on the basis of which criteria, according to which guidelines, insufficient diagnostic imaging was performed, and the medical examination was not precisely described. Assessment on various scales is not sufficient in assessing the patient's clinical condition, assessment on scales is necessary to assess the effectiveness of therapy and publication of results in the literature.

Reply- The use of the SF-36 is a non-symptom specific objective outcome measure and will suffice for reportage of findings for this case report. The examination of the patient was performed by a licensed physician and complied with all state and federal guidelines for the evaluation and treatment of musculoskeletal disorders. The insinuation that this was not the case is an accusation of medical malpractice. MRI was not indicated and would not have been financially viable for this patient.  

  • In the discussion, attention is drawn to the excess of citations, e.g. "The treatment outcomes for the present case were consistent with previously reported case reports [19], cohort and case series [20], randomized controlled trials [21,22,23,24,25] , systematic reviews of the literature and other prior investigations [26,27,28,29]."
  •  

Reply- Thank you for your input.  We feel this will suffice for this simple case report and do not consider the references excessive. And we recommend the reviewer read the references in order to better understand the treatment protocols and their extensive research publication history.

  • Due to methodological irregularities and the lack of precise diagnosis, the presented conclusions are inadequate.

Reply- Thank you for your opinion. The authors disagree.